# Influence of Farm Environment on Asthma during the Life Course: A Population-Based Birth Cohort Study in Northern Finland

**DOI:** 10.3390/ijerph20032128

**Published:** 2023-01-24

**Authors:** Marko T. Kantomaa, Mimmi Tolvanen, Miia Halonen, Cecilie Svanes, Marjo-Riitta Järvelin, Sylvain Sebert

**Affiliations:** 1Research Unit of Population Health, Faculty of Medicine, University of Oulu, 90220 Oulu, Finland; 2Department of Occupational Medicine, Haukeland University Hospital, N-5021 Bergen, Norway; 3Centre for International Health, Department of Global Public Health and Primary Care, University of Bergen, N-5009 Bergen, Norway; 4Unit of Primary Care, Oulu University Hospital, 90220 Oulu, Finland; 5MRC Centre for Environment and Health, Department of Epidemiology and Biostatistics, School of Public Health, Imperial College London, London W2 1PG, UK

**Keywords:** asthma, asthma epidemiology, environmental exposures, farming, life course, early life origins

## Abstract

We investigated the influence of a farming environment on asthma at three time points from birth to 46 years using the Northern Finland Birth Cohort 1966 (n = 10,926). The prevalence of asthma was investigated by postal questionnaires at 14, 31 and 46 years of age. Exposure to a farming environment was assessed by a postal questionnaire at birth and at 31 and 46 years of age. Odds ratios (ORs) and their 95% confidence intervals (95% CIs) for the prevalence of asthma were obtained from multinomial logistic regression, stratified by sex. Being born in a farmer family was potentially causally associated with lower risk of asthma in males at 31 years of age (OR 0.56, 95% CI 0.37, 0.85) and in females at 46 years of age (OR 0.64, 95% CI 0.44, 0.95). Working as a farmer was not associated with asthma. Exposure to a farming environment in childhood may have a lifelong impact on developing asthma from birth through young adulthood and until middle age, indicating that ‘immune deviation’ may persist throughout life.

## 1. Introduction

The prevalence of asthma has increased steeply during recent decades, affecting approximately 20% of the population worldwide [1,2]. This relatively rapid change suggests that the increase may be due to 20th century environmental changes, including extensive urbanisation [3]. Increasing urbanisation may contribute to changes in lifestyle and environmental exposures (e.g., air pollution, smoking and infections) that affect allergic mechanisms and the rising prevalence of asthma [4,5,6]. Many of these changes are associated with early life and lifelong risk factors for the development of asthma [3].

According to the extant literature, being born and raised on a farm appears to protect against asthma [7,8,9,10,11], which may be due to greater or more diverse microbial exposure in the farming environment. Although controversial, it has been suggested that the relationship between farm exposure and asthma may be explained by the ‘hygiene hypothesis’, which proposes that the development of asthma, especially atopic asthma and allergy, may be prevented via prenatal and/or early childhood exposure to immune system stimulants, e.g., bacteria, viruses and endotoxins [11,12,13].

Studies among adult farmers have suggested that protection against asthma may continue into adulthood [14,15,16,17]. However, not all studies have reported that farming provides a protective effect in adults, and it is unclear whether the reduced risk of asthma in farmers is due to early childhood exposure, current exposure, or a combination of both. Furthermore, the protective effect of a farming environment may vary according to demographic characteristics and clinical phenotypes of asthma. For example, a population-based (RHINE) study indicated that the urban–rural gradient was evident only among women and smokers and only for the late-onset asthma phenotype [8].

It is increasingly recognised that the timing, load, and route of allergen exposure affect allergic disease development [3,11]. For example, in utero exposure may contribute to the low prevalence of asthma in farmers’ children, but long-term exposure may be needed to maintain protection [18]. The evidence concerning potential lifelong effects of childhood farm exposure on asthma during the life course is still scarce and inconsistent. Using the Northern Finland Birth Cohort 1966 (University of Oulu, 1966), we aimed to investigate the influence of a farming environment on asthma from birth to 46 years of age. We hypothesised that exposure to a farming environment in childhood protects against asthma from birth to middle age.

## 2. Materials and Methods

This study is based on data from the Northern Finland Birth Cohort 1966 (NFBC1966). The original study population included 12,058 alive births to mothers in the two northernmost provinces of Finland. Pregnancies were followed prospectively from the first antenatal contact (10–16 weeks), and the offspring were then examined at birth and 1, 7, 14, 31 and 46 years either by questionnaires and/or clinical examinations. At these time points, a wide range of health, lifestyle, demographic and socioeconomic data were gathered using questionnaires and clinical examinations. The analysis in the present study included 10,926 participants (5512 males and 5414 females) who had valid data on asthma and childhood farm exposure (being born in a farmer family or not). NFBC1966 conformed to the principles of the Declaration of Helsinki. The participants took part voluntarily and signed informed consent forms. The Ethical Committee of the Northern Ostrobothnia Hospital District approved the study protocol.

*Asthma* was obtained from the questionnaires and defined as an affirmative answer to either ‘Do you have or have you ever had asthma?’ or ‘Have you ever had asthma diagnosed by a doctor?’ and a retrospectively reported age of onset.

Information on being born in a *farmer family* was obtained in the parental questionnaire by asking about the mother’s occupational socioeconomic position (SEP) when the child was born. Children born to mothers reporting their occupational SEP as a ‘farmer’ or ‘farmer’s wife’ were categorised as *children born in a farmer family*.

Potential confounding variables included sex, living place, SEP, birthweight, family size, maternal smoking and body mass index (BMI) at birth (parental questionnaire); BMI, physical activity and father’s smoking at age 14 (parental and adolescent questionnaire); and occupation (farmer), SEP, BMI, physical activity and diet at the age of 31 (postal questionnaire).

Potential confounders measured at birth: Living place was defined as the municipality where the family lived at the time the child was born. The variable was categorised as: (1) city/urban and (2) rural. As a measure of SEP, a factor score was created based on three variables: mother’s and father’s occupations categorised as (1) professional, (2) skilled worker/farmer or (3) unskilled worker and a variable identifying farmer families [19]. In the factor scores, a smaller value indicated a higher SEP. Family size was measured by asking: ‘How many people belong to the household?’ Maternal smoking status was based on responses to the question: ‘During the 12 months preceding the pregnancy, did the mother smoke at least one cigarette or one pipeful of tobacco a day?’ The response categories were (1) no and (2) yes. Maternal BMI was calculated as the individual’s self-reported weight divided by the square of the height (kg/m^2^).

Potential confounders measured in adolescence: Adolescent BMI was calculated on the basis of self-reported height and weight. Frequency of physical activity was measured by asking: ‘How often are you involved in one or more sports outside school?’ The response categories were (1) every day, (2) every second day, (3) twice a week, (4) once a week, (5) every second week, (6) once a month and (7) usually never. Father’s smoking was measured by adolescent questionnaire. The response categories were (1) never, (2) sometimes, but not anymore, (3) he smokes and (4) I don’t know.

Potential confounders measured in adulthood: As a measure of SEP, a factor score was created based on three variables: occupational level ((1) upper-level employees, (2) lower-level employees/entrepreneurs, (3) manual workers/farmers or (4) not working) and identification variables for entrepreneurs and farmers [19]. In the factor scores, a smaller value indicated a higher SEP. BMI was calculated on the basis of a clinical examination (postal questionnaire if clinical examination missing). Physical activity was self-reported by answering the question: ‘How often do you exercise in your leisure time?’ The response categories were (1) every day, (2) every second day, (3) twice a week, (4) once a week, (5) every second week, (6) once a month and (7) usually never. Allergic sensitisation was measured by skin prick tests to assess sensitivity to three of the most common allergens in Finland, i.e., cat, birch and timothy, and sensitivity to the house dust mite (*Dermatophagoides pteronyssinus*) [20] The variable was categorised as (1) mono/non-sensitised and (2) polysensitised. 

Consumption of food and beverages was surveyed using a 32-item food frequency questionnaire. The participants were asked to consider their habitual food consumption during the previous 6 months. Items describing a healthy diet were frequent consumption of plain dairy yogurts, rye bread/crispbread, porridge, salad dressings, fresh vegetables, cooked vegetables, fruits, fresh or frozen berries and fish (9 food items). Items describing an unhealthy diet were frequent consumption of sausages/frankfurters, cold cuts, fried potatoes/French fries, sugar-sweetened soft drinks, white bread and hamburgers and pizzas (6 food or drink items). A value of either zero (less frequent consumption) or one point (more frequent consumption) was assigned to each item, and sum scores for healthy and unhealthy diets were calculated [21].

Statistical analyses: Sample characteristics were summarised descriptively, using mean and SD values for continuous data and frequencies and percentages for categorical data. The longitudinal associations of being born in a farmer family with the prevalence of asthma were examined via multinomial logistic regression analysis, stratified by sex. The results of the regression analyses are presented with standardised regression coefficients and 95% confidence intervals (95% CIs). The analyses were adjusted in the multivariable models as follows (asthma at 14 years of age as an outcome): Model 1: living place at birth, SEP, birth weight, family size at birth, maternal smoking and maternal BMI; Model 2: adding BMI, physical activity and father’s smoking at the age of 14. In the models with asthma in adulthood (31 or 46 years) as an outcome, Model 2 was further adjusted for occupation (farmer), SEP, BMI, physical activity and diet at 31 years of age. In the present analysis, only singleton, term-born participants were included in the analysis. In addition, there were some missing values in confounding factors, which were not included in the final analyses. The statistical analyses were conducted in 2020 using SPSS for Windows 19.0.

## 3. Results

The sex-specific distributions for the basic characteristics of the study population (n = 10,926) are shown in Table 1. The prevalence of asthma was 1.8% at 14, 12% at 31 and 14% at 46 years of age. Females reported slightly less asthma than males at age 14 and more asthma compared to males at age 46 (Table 1). About 19% of the children were born in a farmer family. At the age of 31, almost 3% of the participants were working as farmers, males more commonly than females (Table 1). Among the 575 participants who reported having asthma at 31 years of age, 370 (64%) reported having asthma also at age 46 (Table 2). Information on atopic asthma (skin prick test) was available on ca. 5000 participants. Among males, 1394 of those reporting asthma at age 31 years were mono/non-sensitised compared to 244 males who were polysensitised. Among females, 1524 of those reporting asthma at age 31 years were mono/non-sensitised compared to 260 females who were polysensitised.

Being born in a farmer family was not statistically significantly associated with asthma at age 14 years (Table 3). However, males born in farmer families (OR 0.56, 95% CI 0.37, 0.85) were less likely to have asthma at age 31 years compared to those born in nonfarmer families after adjustment for potential confounders at birth and at 14 and 31 years of age (Table 4). In females, being born in a farmer family was not statistically significantly associated with asthma at age 31 (Table 4).

At the age of 46 years, females born in farmer families (OR 0.64, 95% CI 0.44, 0.95) were less likely to have asthma compared to those born in nonfarmer families after adjustment for potential confounders at birth and at 14 and 31 years of age (Table 5). In males, being born in a farmer family was not statistically significantly associated with asthma at the age of 46 years (Table 5).

At 31 and 46 years of age, female farmers were more likely to have asthma compared to nonfarmers. However, the associations were not statistically significant and did not alter the associations between early childhood farm exposure and asthma later in life (Table 4 and Table 5). The fully adjusted models at 14, 31 and 46 years of age explained 1.3–3.3% of the variance in asthma, as indicated by the R^2^ values (Table 3, Table 4 and Table 5). As sensitivity analyses, we also tested for the potential modifying effect of allergic sensitisation on the association between childhood farming environment and asthma later in life, which indicated no evidence of effect modification.

## 4. Discussion

In this population-based longitudinal study, children born in farmer families had significantly less asthma in young adulthood (males) and in middle age (females) but not in adolescence when compared to children born in nonfarmer families. These results were found after adjusting for several potential confounders at each time point, including smoking, SEP, diet and physical activity. The reduced risk was consistent for both adult age groups from 31 to 46 years of age, indicating that exposure to a farming environment in childhood may have a lifelong impact on developing asthma from birth through young adulthood and until middle age.

To the best of our knowledge, this is the first study to investigate the life course effects of a farming environment on asthma at three time points from birth to 46 years of age. Our findings are comparable to current evidence: in three large European cohort studies, exposure to a farming environment in childhood was found to protect against asthma and asthma-like symptoms [22]. However, inconsistent results have also been reported [23], potentially explained by cohort effects [7,24] and different farm locations and farming practices within Europe.

According to previous studies, in utero exposure may contribute to the low prevalence of allergic diseases in farmers’ children, but both prenatal and early childhood exposure may be required for optimal protection [18]. Farming is known to be associated with increased exposures to bacterial endotoxin and other microbial agents [11,25], which may inhibit T-helper type 2 cell immune responses and the subsequent development of T-helper type 2-dependent diseases, including atopic asthma and allergy [12,13,26]. Our results on life course models indicate that early life farm exposure has protective effects on asthma in young adulthood and in middle age, supporting studies reporting that potential ‘immune deviation’ induced by the farm environment may take place throughout life [11,25,27]. However, further investigation of the life course effects is clearly needed, including the role of sensitisation to aeroallergens in the farming effect [28].

Our results did not reveal significant differences on sex-specific effects, notwithstanding that some previous studies have showed a somewhat stronger protective effect for females than for males [8,29]. The explanation for the sex-specific findings remains unknown, but it has been suggested that girls are more frequently in contact with animals, such as horses and stables [8]. However, in line with the present results, a Danish study reported no sex-related differences in asthma risk among farming students [30]. 

Interestingly, at the ages of 31 and 46 years, female farmers were more likely to have asthma compared to nonfarmers, but the association was not statistically significant. It should be noted that our analysis was not designed to specifically address this topic and that the number of farmers in our study population was limited. According to Omland et al. [30], protective effects may be limited to early life exposure only, as occupational farm exposure later in life may increase the risk of asthma. Farm environments may also reflect non-exposure to several risk factors for asthma such as smoking and air pollution [8,31], as well as high endotoxin levels indoors [9,32,33] typical to urban environments. The heterogeneity of farming exposures as well as other adult exposures, especially occupational factors, may also explain the inconsistent relationship between farming and asthma in adulthood [8].

However, regarding the present results, it is important to note that since the information on farming environment was collected in 1966, most of the participants have moved away from their birthplace due to heavy rural-to-urban migration in Finland during the 1960s and 1970s [34]. Furthermore, at the time of the data collection in 1966, in Finland, farm animal ownership was typically not restricted to professional farmers but was more common than it is in the 21st century [20]. Currently, 70% of farms in Finland have crop production as the production line, 25% of farms are livestock farms and 13.5% of the farming area is occupied by organic farming (data were sourced from the The Natural Resources Institute Finland: www.luke.fi). Due to hard frosts, which control plant diseases and kill pests, use of chemical plant protection products in Finland is lower than in the rest of the Europe (www.luke.fi (accessed on 15 December 2022)) [35].

An important strength of this study is the prospective population-based study setting, which provides robust information concerning the effects of early childhood farm exposure on asthma later in life that is generalisable to a general population. Furthermore, we had access to rich questionnaire and clinical data at each time point and were able to adjust properly for a large number of important potentially confounding factors. One limitation of the present study is the fact that information on all variables of interest was self-reported, and the question in adulthood did not specifically ask the timeline as to whether the development occurred during childhood or adulthood—as a result, recall bias and misclassification are possible. However, both self-reported and doctor-diagnosed measures seem to have a high specificity, although sensitivity is low [36]. Furthermore, in the present study, an analysis of changes in the prevalence of asthma from 31 to 46 years of age showed that 64% of participants who reported ever having asthma at the age of 31 years reported ever having asthma at age 46. In the present study, we had limited information on the phenotype of asthma, restricting evaluation of the magnitude of a potential protective effect of farm exposure on asthma. The study also lacks information on different farming practices (e.g., with or without livestock), which may have an impact on the risk of developing asthma. Furthermore, the lack of measures of environmental exposures as well as breast feeding—which may be considered potential confounders—is a limitation of our study.

## 5. Conclusions

The potential protective effects of farm exposure in early childhood against asthma persisted in males in young adulthood and in females in mid-age adulthood, indicating that ‘immune deviation’ may persist throughout life. This may help when designing preventive actions focusing on the microbial environment in early childhood and extensive urbanisation.

## Figures and Tables

**Table 1 ijerph-20-02128-t001:** Characteristics of the participants in the Northern Finland Birth Cohort 1966, 1966–2012.

Characteristics	Male (n = 5512)		Female (n = 5414)		All (n = 10,926)	
	n	%, Mean (SD)	n	%, Mean (SD)	n	%, Mean (SD)
**At birth**						
Farmer family ^1^	5510		5412		10,923	
No	4519	82.0	4385	81.0	8904	81.5
Yes	991	18.0	1028	19.0	2019	18.5
Living place						
City/urban	1647	29.9	1651	30.5	3306	30.2
Rural	3865	70.1	3763	69.5	7641	69.8
SEP ^2^	5510	−0.2 (0.999)	5413	−0.02 (0.999)	10,923	−0.02 (0.99)
Birth weight 100 g	5511	35.5 (5.4)	5414	34.2 (5.1)	10,925	34.9 (5.3)
Family size (number of persons)	5411	4.3 (2.1)	5306	4.3 (2.1)	10,717	4.3 (2.1)
Maternal smoking ^3^	5389		5293		10,682	
Yes	1174	21.3	1113	21.0	2287	21.4
No	4215	78.2	4180	79.0	8395	78.6
Maternal BMI	4996	23.1 (3.2)	4961	23.1 (3.2)	9957	23 (3.2)
**At age 14**						
Asthma ^4^	5512		5414		10,926	
Never	5394	97.9	5340	98.6	10,734	98.2
Ever	118	2.1	74	1.4	192	1.8
BMI	5066	19.3 (2.6)	5046	19.4 (2.5)	10,112	19.6 (2.5)
Frequency of physical activity	5393		5314		10,707	
Every day	1220	22.6	650	12.2	1870	17.5
Every second day	1362	25.3	809	15.2	2171	20.3
Twice a week	1180	21.9	1164	21.9	2344	21.9
Once a week	671	12.4	1026	19.3	1697	15.8
Every second week	139	2.6	196	3.7	335	3.1
Once a month	152	2.8	238	4.5	390	3.6
Usually never	669	12.4	1231	23.2	1900	17.7
Father’s smoking	5232		5130		10,362	
No	1283	24.5	1278	24.9	2561	24.7
Yes	3949	75.5	3852	75.1	7801	75.3
**At age 31**						
Asthma ^4^	3916		4337		8253	
Never	3468	88.6	3832	88.4	7300	88.5
Ever	448	11.4	505	11.6	953	11.5
Farmer	3908		4339		8247	
Yes	143	3.7	97	2.2	240	2.9
No	3765	96.3	4242	97.8	8007	97.1
SEP ^5^	3908	−0.0 (1.1)	4339	−0.01 (0.897)		
BMI	3936	25.3 (3.6)	4315	23.9 (4.5)	8251	24.5 (4.1)
Physical activity	3917		4324		8241	
Once a month or less	848	21.6	816	18.9	1664	20.2
2–3 times per month	537	13.7	587	13.6	1124	13.6
Once a week	832	21.2	1083	25.0	1915	23.2
2–3 times per week	1148	29.3	1311	30.3	2459	29.8
4–6 times per week	452	11.5	410	9.5	862	10.5
Daily	100	2.6	117	2.7	217	2.6
Healthy diet sum score ^6^	3831	2.6 (1.7)	4263	3.6 (1.8)	8094	3.1 (1.8)
Unhealthy diet sum score ^7^	3784	2.5 (1.5)	4224	1.6 (1.2)	8008	2.0 (1.4)
**At age 46**						
Asthma ^4^	2883		3471		6593	
Never	2536	88.0	2927	84.3	5672	86.0
Ever	347	12.0	544	15.7	921	14.0

BMI, body mass index; SEP, socioeconomic position. ^1^ Children born to mothers reporting their occupational SEP as a ‘farmer’ or ‘farmer’s wife’ were categorised as children born in a farmer family. ^2^ SEP factor score was based on three variables: mother’s and father’s occupations categorised as (1) professional, (2) skilled worker/farmer or (3) unskilled worker and a variable identifying farmer families. A smaller value indicated a higher SEP. ^3^ Mothers who reported smoking at least one cigarette or one pipeful of tobacco a day during the 12 months preceding the pregnancy way categorised as ‘smoking’. ^4^ Self-reported or doctor-diagnosed asthma (current or ever) obtained from the questionnaires. ^5^ SEP factor score was based on three variables: occupational level (categorised as (1) upper-level employees, (2) lower-level employees/entrepreneurs, (3) manual workers/farmers or (4) not working) and identification variables for entrepreneurs and farmers. A smaller value indicated a higher SEP. ^6^ Sum of healthy diet choices, scale 0–9, the bigger the better. ^7^ Sum of unhealthy diet choices, scale 0–6, the smaller the better.

**Table 2 ijerph-20-02128-t002:** Changes in the prevalence of asthma from 31 to 46 years of age in the Northern Finland Birth Cohort 1966, 1997–2012.

	Asthma ^1^ at Age 46		
Asthma ^1^ at Age 31	Never (%)	Ever (%)	Total (%)
Females			
Never	2566 (89.9)	287 (10.1)	2853 (100.0)
Ever	105 (33.1)	212 (66.9)	317 (100.0)
Total	2671 (84.3)	499 (15.7)	3170 (100.0)
Males			
Never	2116 (93.6)	145 (6.4)	2261 (100.0)
Ever	100 (38.8)	158 (61.2)	258 (100.0)
Total	2216 (88.0)	303 (12.0)	2519 (100.0)

^1^ Self-reported or doctor-diagnosed asthma (current or ever) obtained from the questionnaires.

**Table 3 ijerph-20-02128-t003:** Multivariable regression analysis of farm environment and asthma at age 14 years in the Northern Finland Birth Cohort 1966, 1966–1980.

	Asthma ^1^							
	Females				Males			
	Model 1 (n = 4810)		Model 2 (n = 4206)		Model 1 (n = 4849)		Model 2 (n = 4197)	
	OR	95% CI	OR	95% CI	OR	95% CI	OR	95% CI
**Birth**								
Farmer family ^2^	0.530	0.184, 1.524	0.549	0.189, 1.590	0.818	0.412, 1.627	0.855	0.398, 1.840
Living place	0.642	0.386, 1.067	0.680	0.400, 1.155	0.706	0.462, 1.081	0.843	0.525, 1.352
SEP ^3^	0.978	0.749, 1.279	0.950	0.718, 1.257	0.958	0.769, 1.193	0.968	0.758, 1.238
Birth weight	1.024	0.974, 1.076	1.018	0.967, 1.072	1.005	0.969, 1.043	0.999	0.959, 1.041
Family size	0.889	0.763, 1.037	0.905	0.772, 1.060	1.001	0.897, 1.118	1.009	0.892, 1.140
Maternal smoking ^4^	1.278	0.739, 2.210	1.348	0.759, 4.448	1.188	0.760, 1.858	1.249	0.767, 2.035
Maternal BMI	0.997	0.916, 1.084	0.992	0.906, 1.086	0.990	0.927, 1.057	0.932	0.861, 1.010
**14 years**								
BMI			1.009	0.910, 1.117			1.043	0.964, 1.128
Physical activity ^5^			1.020	0.905, 1.151			1.047	0.939, 1.167
Paternal smoking ^6^			1.140	0.628, 2.069			1.404	0.815, 2.417
R^2^	0.024		0.021		0.006		0.013	

BMI, body mass index; CI, confidence interval; OR, odds ratio; SEP, socioeconomic position. ^1^ Self-reported or doctor-diagnosed asthma (current or ever) obtained from the questionnaires. ^2^ Children born to mothers reporting their occupational SEP as a ‘farmer’ or ‘farmer’s wife’ were categorised as children born in a farmer family. ^3^ SEP factor score was based on three variables: mother’s and father’s occupations categorised as (1) professional, (2) skilled worker/farmer or (3) unskilled worker and a variable identifying farmer families. A smaller value indicated a higher SEP. ^4^ Mothers who reported smoking at least one cigarette or one pipeful of tobacco a day during the 12 months preceding the pregnancy were categorised as ‘smoking’. ^5^ Frequency of sports outside school (‘once a month or less’–‘daily’). ^6^ Participants who reported smoking on 1–7 days a week were classified as ‘smoking’.

**Table 4 ijerph-20-02128-t004:** Regression analysis of farm environment and asthma at age 31 years in the Northern Finland Birth Cohort 1966, 1966–1997.

	Asthma ^1^							
	Females				Males			
	Model 1 (n = 3859)		Model 2 (n = 3188)		Model 1 (n = 3465)		Model 2 (n = 2826)	
	OR	95% CI	OR	95% CI	OR	95% CI	OR	95% CI
**Birth**								
Farmer family ^2^	0.757	0.539, 1.064	0.773	0.534, 1.119	0.704	0.492, 1.007	0.557	0.365, 0.851
Living place	0.855	0.681, 1.073	0.873	0.679, 1.123	0.803	0.632, 1.021	0.906	0.690, 1.190
SEP ^3^	0.894	0.795, 1.005	0.875	0.767, 0.997	0.926	0.820, 1.046	0.844	0.734, 0.971
Birth weight	1.002	0.981, 1.022	0.998	0.976, 1.021	0.998	0.978, 1.018	0.996	0.973, 1.019
Family size	1.028	0.973, 1.086	1.043	0.981, 1.108	1.600	1.000, 1.123	1.078	1.009, 1.151
Maternal smoking ^4^	1.302	1.022, 1.658	1.274	0.967, 1.677	1.094	0.846, 1.416	1.075	0.801, 1.443
Maternal BMI	0.982	0.948, 1.016	0.962	0.925, 1.001	1.009	0.975, 1.045	0.993	0.952, 1.035
**14 years**								
BMI			1.000	0.948, 1.056			0.992	0.938, 1.048
Physical activity ^5^			1.026	0.971, 1.084			1.028	0.964, 1.097
Paternal smoking ^6^			0.849	0.664, 1.087			1.227	0.920, 1.634
**31 years**								
Farmer			1.141	0.465, 2.801			0.702	0.296, 1.661
SEP ^7^			0.960	0.829, 1.112			1.079	0.938, 1.241
BMI			1.060	1.030, 1.090			1.037	0.999, 1.077
Physical activity ^8^			1.117	1.027, 1.216			1.079	0.987, 1.179
Healthy diet ^9^			0.947	0.890, 1.009			0.957	0.888, 1.031
Unhealthy diet ^10^			1.006	0.921, 1.099			0.998	0.921, 1.082
R^2^	0.007		0.026		0.006		0.017	

BMI, body mass index; CI, confidence interval; OR, odds ratio; SEP, socioeconomic position. ^1^ Self-reported or doctor-diagnosed asthma (current or ever) obtained from the questionnaires. ^2^ Children born to mothers reporting their occupational SEP as a ‘farmer’ or ‘farmer’s wife’ were categorised as children born in a farmer family. ^3^ SEP factor score was based on three variables: mother’s and father’s occupations categorised as (1) professional, (2) skilled worker/farmer or (3) unskilled worker and a variable identifying farmer families. A smaller value indicated a higher SEP. ^4^ Mothers who reported smoking at least one cigarette or one pipeful of tobacco a day during the 12 months preceding the pregnancy were categorised as ‘smoking’. ^5^ Frequency of sports outside school (‘once a month or less’–‘daily’). ^6^ Participants who reported smoking on 1–7 days a week were classified as ‘smoking’. ^7^ SEP factor score was based on three variables: occupational level (categorised as (1) upper-level employees, (2) lower-level employees/entrepreneurs, (3) manual workers/farmers or (4) not working) and identification variables for entrepreneurs and farmers. A smaller value indicated a higher SEP. ^8^ Frequency of exercise during leisure time (‘every day’–‘usually never’). ^9^ Sum of healthy diet choices, scale 0–9, the bigger the better. ^10^ Sum of unhealthy diet choices, scale 0–6, the smaller the better.

**Table 5 ijerph-20-02128-t005:** Regression analysis of farm environment and asthma at age 46 years in the Northern Finland Birth Cohort 1966, 1966–2012.

	Asthma ^1^							
	Females				Males			
	Model 1 (n = 3106)		Model 2 (n = 2389)		Model 1 (n = 2569)		Model 2 (n = 1861)	
	OR	95% CI	OR	95% CI	OR	95% CI	OR	95% CI
**Birth**								
Farmer family ^2^	0.805	0.573, 1.130	0.643	0.436, 0.948	0.762	0.514, 1.129	0.819	0.509, 1.319
Living place	0.833	0.668, 1.040	0.883	0.685, 1.138	0.934	0.707, 1232	0.886	0.635, 1.238
SEP ^3^	1.020	0.907, 1.147	0.985	0.860, 1.128	0.901	0.785, 1.035	0.869	0.733, 1.030
Birth weight	0.996	0.976, 1.016	0.996	0.972, 1.020	0.986	0.964, 1.008	0.972	0.945, 1.000
Family size	1.026	0.972, 1.084	1.053	0.989, 1.121	1.014	0.948, 1.084	1.067	0.986, 1.154
Maternal smoking ^4^	1.323	1.042, 1.681	1.189	0.893, 1.582	1.253	0.934, 1.680	1.185	0.827, 1.700
Maternal BMI	0.994	0.961, 1.027	0.965	0.927, 1.004	1.053	1.013, 1.094	1.043	0.994, 1.094
**14 years**								
BMI			1.014	0.959, 1.071			0.990	0.924, 1.061
Physical activity ^5^			0.994	0.939, 1.053			1.034	0.956, 1.118
Paternal smoking ^6^			1.150	0.885, 1.496			1.246	0.886, 1.753
**31 years**								
Farmer			1.666	0.740, 3.754			0.519	0.172, 1.567
SEP ^7^			1.031	0.884, 1.203			1.009	0.856, 1.188
BMI			1.057	1.027, 1.089			1.032	0.982, 1.084
Physical activity ^8^			0.983	0.902, 1.071			1.039	0.931, 1.159
Healthy diet ^9^			0.974	0.913, 1.038			1.047	0.960, 1.142
Unhealthy diet ^10^			1.042	0.951, 1.141			0.954	0.865, 1.053
R^2^	0.008		0.033		0.009		0.022	

BMI, body mass index; CI, confidence interval; OR, odds ratio; SEP, socioeconomic position. ^1^ Self-reported or doctor-diagnosed asthma (current or ever) obtained from the questionnaires. ^2^ Children born to mothers reporting their occupational SEP as a ‘farmer’ or ‘farmer’s wife’ were categorised as children born in a farmer family. ^3^ SEP factor score was based on three variables: mother’s and father’s occupations categorised as (1) professional, (2) skilled worker/farmer or (3) unskilled worker and a variable identifying farmer families. A smaller value indicated a higher SEP. ^4^ Mothers who reported smoking at least one cigarette or one pipeful of tobacco a day during the 12 months preceding the pregnancy were categorised as ‘smoking’. ^5^ Frequency of sports outside school (‘once a month or less’–‘daily’). ^6^ Participants who reported smoking on 1–7 days a week were classified as ‘smoking’. ^7^ SEP factor score was based on three variables: occupational level (categorised as (1) upper-level employees, (2) lower-level employees/entrepreneurs, (3) manual workers/farmers or (4) not working) and identification variables for entrepreneurs and farmers. A smaller value indicated a higher SEP. ^8^ Frequency of exercise during leisure time (‘every day’–‘usually never’). ^9^ Sum of healthy diet choices, scale 0–9, the bigger the better. ^10^ Sum of unhealthy diet choices, scale 0–6, the smaller the better.

## Data Availability

NFBC data are available from the University of Oulu, Infrastructure for Population Studies. Permission to use the data can be requested for research purposes via an electronic material request portal. In the use of data, we follow the EU general data protection regulation (679/2016) and Finnish Data Protection Act. The use of personal data is based on the cohort participant’s written informed consent at their latest follow-up study, which may cause limitations to their use. Please contact the NFBC project center (NFBCprojectcenter@oulu.fi) and visit the cohort website (www.oulu.fi/nfbc (accessed on 15 December 2022)) for more information.

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
