# Peer review of "Influence of Farm Environment on Asthma during the Life Course: A Population-Based Birth Cohort Study in Northern Finland"

_ijerph, 2023, doi:10.3390/ijerph20032128_

Round 1

Reviewer 1 Report

The manuscript is definitely worth publishing. It is of significant scientific interest. The complex and not completely clarified connection between environmental factors and the etiopathogenesis of asthma, including the controversial "hygiene hypothesis", is a topic of great interest. I believe the manuscript will be interesting to the readers of the journal. It can be published in the present form except that I have one really minor comment:

·         The subjects' features of allergic sensitization, which were reported in the Results section, were not specifically mentioned in the Materials and Methods section. The reader of the manuscript can only assume that the variable related to allergic sensitization probably falls under the phrase "...a wide range of health, lifestyle, demographic and socioeconomic data were gathered using questionnaires and clinical examinations"...  In my oppinion, the methods of allergic evaluation of the subjects should be specifically stated in the Materials and Methods section.

Author Response

Point 1: The subjects' features of allergic sensitization, which were reported in the Results section, were not specifically mentioned in the Materials and Methods section. The reader of the manuscript can only assume that the variable related to allergic sensitization probably falls under the phrase "...a wide range of health, lifestyle, demographic and socioeconomic data were gathered using questionnaires and clinical examinations"...  In my oppinion, the methods of allergic evaluation of the subjects should be specifically stated in the Materials and Methods section.

Response 1: Thank you for this comment. We have now added information on the methods of allergic evaluation in the Materials and Methods section (lines 117–120) as follows:

Allergic sensitisation was measured by skin prick tests to assess sensitivity to three of the most common allergens in Finland, i.e. cat, birch and timothy, and sensitivity to the house dust mite (Dermatophagoides pteronyssinus) [20]. The variable was categorised as: 1) mono/non-sensitised and 2) polysensitised. 

Reviewer 2 Report

Excellent article on relation of farming exposure in life course with asthma. I was especially interested in the fact that living in "farming" environment might protect against asthma and working as a farmer at older age might have opposite effect!

Few concerns from my part:

1. How was consent taken? As I understand: at the beginning it was obtained from mother on child birth. However, at what point was the consent signed later, when the participant grew?

2. Skin tests were available on 5000 participants. Do you have any conclusions that this described relation between asthma and farming environment works on asthma subgroups: atopic, nonatopic, Type2, eosinophillic etc? Such description would be interesting.

Author Response

Point 1: How was consent taken? As I understand: at the beginning it was obtained from mother on child birth. However, at what point was the consent signed later, when the participant grew?

Response 1: Thank you for this comment. In NFBC1966, the written informed consent was obtained from all participants involved in the follow-up studies at 31 and 46 years of age. The use of data is based on participants’ written informed consent at their latest follow-up study. We have now added information on informed consent in the Informed Consent Statement (lines 392–394) as follows:

Informed consent was obtained from all participants involved in the study. The use of data is based on participants’ written informed consent at the latest follow-up study.

Point 2: Skin tests were available on 5000 participants. Do you have any conclusions that this described relation between asthma and farming environment works on asthma subgroups: atopic, nonatopic, Type2, eosinophillic etc? Such description would be interesting.

Response 2: As sensitivity analyses, we tested for potential modifying effect of allergic sensitisation on the association between childhood farming environment and asthma later in life, which indicated no evidence of effect modification. That is, in the present study, the described association between farming environment and asthma did not differ according to the level of allergic sensitisation. Unfortunately, our data do not allow further conclusions according to other asthma subgroups. We have indicated this in the Results section (lines 269–272) as follows:

As sensitivity analyses, we also tested for potential modifying effect of allergic sensitisation on the association between childhood farming environment and asthma later in life, which indicated no evidence of effect modification.